# Applications of statistical experimental designs to improve statistical inference in weed management

**Steven B. Kim[1], Dong Sub Kim[2,3]\*, Christina Magana-Ramirez[1]**

**1** Department of Mathematics and Statistics, California State University, Monterey Bay, Seaside, California, United States of America, **2** Department of Plant Sciences, University of California, Davis, Salinas, California, United States of America, **3** Department of Horticulture, Kongju National University, Yesan, South Korea

\* dongsub@kongju.ac.kr

**Data Availability Statement:** All relevant data are within the manuscript and its Supporting information files.

**Funding:** The authors received no specific funding for this work.

## Abstract

In a balanced design, researchers allocate the same number of units across all treatment groups. It has been believed as a rule of thumb among some researchers in agriculture. Sometimes, an unbalanced design outperforms a balanced design. Given a specific parameter of interest, researchers can design an experiment by unevenly distributing experimental units to increase statistical information about the parameter of interest. An additional way of improving an experiment is an adaptive design (e.g., spending the total sample size in multiple steps). It is helpful to have some knowledge about the parameter of interest to design an experiment. In the initial phase of an experiment, a researcher may spend a portion of the total sample size to learn about the parameter of interest. In the later phase, the remaining portion of the sample size can be distributed in order to gain more information about the parameter of interest. Though such ideas have existed in statistical literature, they have not been applied broadly in agricultural studies. In this article, we used simulations to demonstrate the superiority of the experimental designs over the balanced designs under three practical situations: comparing two groups, studying a dose-response relationship with right-censored data, and studying a synergetic effect of two treatments. The simulations showed that an objective-specific design provides smaller error in parameter estimation and higher statistical power in hypothesis testing when compared to a balanced design. We also conducted an adaptive experimental design applied to a dose-response study with right-censored data to quantify the effect of ethanol on weed control. Retrospective simulations supported the benefit of this adaptive design as well. All researchers face different practical situations, and appropriate experimental designs will help utilize available resources efficiently.

## 1. Introduction

A successful weed management is a key to improve the crop productivity and quality. Researchers have used various response variables in weed control studies such as the viability

**Competing interests:** The authors have declared that no competing interests exist.

of weed seeds [1, 2], the germination of weed seeds [3, 4], the weed emergence [5, 6], the weed density per unit area using hand count [7, 8], and the proportion of area covered by green colors [9, 10]. To identify the effect of treatments for weed control, the analysis of variance (ANOVA) and similar statistical methods have been used. Traditionally, if an ANOVA test rejects the null hypothesis (that all group means are equal), the Tukey's test (also known as the Tukey's honestly significant difference) is often used to determine significantly different groups [11, 12]. The Duncan's multiple range test and the Fisher's least square difference test seem to be alternate choices among researchers [13–19]. These statistical tests assume that data are observed from normal distributions (normality assumption) with equal variance (homogeneity assumption). The normality assumption may not be a big deal in large-sample studies, but large sample sizes cannot resolve the issue of unequal variances. The Duncan test does not control the family-wise rate of Type I Error [20]. In other words, the probability of falsely claiming a difference between any two treatments increases as the number of treatment groups increase, so it should not be recommended when there are many treatments to be compared. To guard against the inflated rate of Type I Error in the comparison of multiple treatments, the Tukey's test or a correction method for multiple testing should be considered.

Balanced designs are commonly used in agricultural experiments [21–23]. A balanced design mitigates the violation of homogeneity assumption [24]. In addition, if the homogeneity assumption is true, a balanced design increases statistical power of hypothesis testing for comparing groups. In a two-sample t-test with the homogeneity assumption, it can be shown by calculus that the standard error, the square root of $\sigma^2(1/n_1 + 1/n_2)$, is minimized when $n_1 = n_2$ for a fixed total sample size $n_1 + n_2$. A balanced design may end up unbalanced due to unexpected reasons (e.g., incorrect implementation of a treatment, invasion of pests, and missing samples). Losing a few data point during an experiment is unfortunate, but it is often not a big deal unless the original sample size was extremely small. The Tukey-Kramer method adjusts the calculation of standard error to account for unequal sample sizes [25, 26].

In some cases, the homogeneity assumption is not plausible when treatments have different expected outcomes. Count data tend to vary more when the mean is higher, and the standard error for comparing two group means, which is the square root of $\sigma_1^2/n_1 + \sigma_2^2/n_2$, is minimized when $n_2 = n_1\sigma_2^2/\sigma_1^2$. In this case, however, researchers may be uncomfortable to guess $\sigma_1^2$ and $\sigma_2^2$ before collecting data. Furthermore, it is not always possible to increase the sample size, but it is possible to control a maximally available sample size. If researchers believe that $\sigma_1^2$ and $\sigma_2^2$ are substantially different, an adaptive design may be a practical suggestion. For example, a researcher may start with a balanced design by spending a portion of an available (fixed) sample size, then the researcher may spend the remaining portion of the fixed sample size to minimize the standard error based on estimated $\sigma_1^2$ and $\sigma_2^2$. An adaptive design is not limited to two phases. It may consist of two or more phases to improve precision in parameter estimation. Although a large number of phases may improve the precision from statistical perspective, such a long adaptive design may not be feasible in practice. Throughout the article, an adaptive design refers to an experiment which is designed in two phases until drawing a final statistical inference.

Ronald Fisher realized that the randomization is needed in order to satisfy the assumption of independent errors, and he introduced the principles of randomization in his book, Statistical Methods for Research Workers [27, 28]. The randomization is an important component of experimental design to reduce bias in parameter estimation. It is a misconception that a small sample size leads to the bias. In fact, a small sample size is associated with the variance which is another important component in parameter estimation. A lower variance can be attained by increasing the sample size, but researchers often have limited resources, time, and labor

capacity. Given a fixed sample size, we can still lower the variance by choosing an unbalanced design carefully. For example, suppose a researcher assumes that two numeric variables $X$ (e.g., treatment level) and $Y$ (e.g., response) are linearly related as $Y = \beta_0 + \beta_1 X + \epsilon$. Further assume that the researcher has four numeric levels of treatment $x_1 < x_2 < x_3 < x_4$ and can afford a total sample size of $n_1 + n_2 + n_3 + n_4 = 20$, where $n_i$ is the number of experimental units assigned to $x_i$ for $i = 1, 2, 3, 4$. If the parameter of interest is $\beta_1$ which quantifies the linear relationship between $X$ and $Y$, the unbalanced design $(n_1, n_2, n_3, n_4) = (10, 0, 0, 10)$ leads to a smaller variance than the balanced design $(n_1, n_2, n_3, n_4) = (5, 5, 5, 5)$ in the estimation of parameter $\beta_1$. As such, when a researcher has a target parameter to be estimated or tested, the researcher would like to seek an unbalanced design to minimize the variance in parameter estimation. However, an optimal design for $\beta_1$ (which is optimal from the theoretical perspective) may not be recommended in practice because it is optimal only under the strong assumption of linearity. Even if the linearity assumption is plausible as an approximation, weed scientists may be (or should be) interested in an adequate strength of a treatment from a variety of perspectives such as the effectiveness of weed control, the impact on the environment, and the cost. In this regard, finding an adequate concentration (or any quantification of treatment strength) would be an important research objective. In a later section of this article, we demonstrate a statistical model and an experimental design to find such a parameter in terms of delaying weed emergence.

The primary focus of this article is to demonstrate that there are many practical situations in agricultural studies that unbalanced designs are better than balanced designs. Here, "better" means a smaller mean square error (which accounts both bias and variance) in parameter estimation and a greater statistical power in hypothesis testing (while respecting a fixed significance level $\alpha$). In particular, we demonstrate three practical situations when the parameter of interest is the difference in two group means (Section 2), the effective concentration of an active treatment in which the median time to weed emergence is doubled when compared to the control (Sections 3 and 4), and the synergistic effect of two treatments (Section 5).

In addition, by spending an available total sample size in two phases (referred to as an adaptive design in this article), a researcher may improve the precision of parameter estimation by correcting an assumption made prior to the experiment. For instance, a Bayesian optimal design or a locally optimal design requires researcher's guess about model parameters prior to collecting data [29–31]. In practice, it is challenging to specify an informative prior for model parameters, and an informative prior may severely deviate from the truth. In such a case, a researcher may regret for making a decision at once with scarce knowledge. To address the caveat of designing an experiment at once before collecting data, a researcher may use a non-informative prior to allocate a portion of the total sample size, then decide an experimental design for the remaining portion using an updated posterior. In this Bayesian approach, the key idea is to design each phase of experiment by utilizing all available knowledge (Sections 4 and 5). The benefit of adaptive designs will be demonstrated via simulations in the later sections. There are many statistical methods which have been developed to make adaptive decisions in clinical trials [32–35]. The idea of an adaptive design is not new in scientific communities, and in this article, it is discussed in the context of agricultural studies.

## 2. Comparing two groups

### 2.1. Assumptions

We assume that there are two treatments to be compared and further assume that the response variable is generated from normal distributions, $N(\mu_1, \sigma_1^2)$ for the first treatment group (say group 1) and $N(\mu_2, \sigma_2^2)$ for the second treatment group (say group 2). The null hypothesis is

$H_0$: $\mu_1 - \mu_2 = 0$, and the alternative hypothesis is $H_1$: $\mu_1 - \mu_2 \neq 0$. Let $n_1$ and $n_2$ be the sample size for group 1 and group 2, respectively. Consider two-sample t-test without the equal variance assumption, and the degrees of freedom for the T statistic is estimated by the Welch-Satterthwaite equation.

Suppose Researcher 1 performs the t-test using the balanced design with a total sample size of $n_1 + n_2 = 40$, so $n_1 = n_2 = 20$. Among possible choices of $(n_1, n_2) = (2, 38), (3, 37), \ldots, (37, 3), (38, 2)$, the balanced design minimizes the standard error (SE), the square root of $\sigma_1^2/n_1 + \sigma_2^2/n_2$, when $\sigma_1 = \sigma_2$. In general, if the researcher knew the true values of $\sigma_1$ and $\sigma_2$, the optimal choice for minimizing $\sigma_1^2/n_1 + \sigma_2^2/n_2$ would be $n_1 = 40\sigma_1/(\sigma_1 + \sigma_2)$ and $n_2 = 40 - n_1$ rounded.

Suppose Researcher 2 chooses an adaptive design (given the total sample size 40) as follows. Let $n_1'$ and $n_2'$ be the sample size in the first phase for group 1 and group 2, respectively. Let $n_1''$ and $n_2''$ be the respective sample size in the second phase. The researcher initiates a balanced design of $n_1' = n_2' = 10$, estimates $\sigma_1^2$ and $\sigma_2^2$ by the sample variances $S_1^2$ and $S_2^2$, respectively, then decides $n_1''$ and $n_2'' = 20 - n_1''$ which minimize the estimated SE, the square root of $S_1^2/(10 + n_1'') + S_2^2/(10 + n_2'')$, for $n_1'' = 0, 1, \ldots, 20$. In other words, the researcher spends one half of the total sample size 40 to learn about $\sigma_1^2$ and $\sigma_2^2$, then spends the remaining half to reduce the SE in the two-sample t-test.

## 2.2. Simulations

To compare statistical power between the design of Researcher 1 (balanced design) and the design of Researcher 2 (adaptive design), simulations scenarios were set at $\mu_1 = 10$, $\sigma_1 = 10$, $\mu_2 = 10, 20, \ldots, 100$, and $\sigma_2 = 10, 20, 50, 100$ at the significance level of $\alpha = 0.05$. Each scenario was simulated 10,000 times. The simulation process for the balanced design is as follows:

1. Fix the values of $\mu_1, \mu_2, \sigma_1$, and $\sigma_2$.

2. Generate a random sample of size $n_1 = 20$ from $N(\mu_1, \sigma_1^2)$.

3. Generate a random sample of size $n_2 = 20$ from $N(\mu_2, \sigma_2^2)$.

4. Perform the two-sample t-test and calculate the p-value.

5. Repeat Steps 2 to 4 10,000 times.

6. Calculate the proportion of times when p-value $<0.05$ to estimate the probability of concluding $H_1$ (power at the significance level $\alpha = 0.05$).

The simulation process for the adaptive design is as follows:

1. Fix the values of $\mu_1, \mu_2, \sigma_1$, and $\sigma_2$.

2. Generate a random sample of size $n_1' = 10$ from $N(\mu_1, \sigma_1^2)$, and estimate $\sigma_1^2$ by the sample variance $S_1^2$.

3. Generate a random sample of size $n_2' = 10$ from $N(\mu_2, \sigma_2^2)$, and estimate $\sigma_2^2$ by the sample variance $S_2^2$.

4. Let $n_2'' = 20 - n_1''$, evaluate $\widehat{SE} = \sqrt{\frac{S_1^2}{10+n_1''} + \frac{S_2^2}{10+n_2''}}$ for $n_1'' = 0, 1, \ldots, 20$, and choose the value of $n_1''$ which minimizes $\widehat{SE}$.

5. Given the chosen value of $n_1''$, generate a random sample of size $n_1''$ from $N(\mu_1, \sigma_1^2)$.

6. Given $n_2'' = 20 - n_1''$, generate a random sample of size $n_2''$ from $N(\mu_2, \sigma_2^2)$.

7. Combine the random samples in Steps 2, 3, 5, and 6 to perform the two-sample t-test and calculate the p-value.

8. Repeat Steps 2 to 7 10,000 times.

9. Calculate the proportion of times when p-value <0.05 to estimate the probability of concluding $H_1$ (power at $\alpha = 0.05$).

All computational work was performed in R Version 4.0.2 [36], and all codes were written by the authors with built-in functions like `t.test`. The simulation results are graphically shown in Fig 1. There is no meaningful difference between the two designs when $\sigma_1 = \sigma_2$ (the upper left panel of Fig 1). The adaptive design provides greater statistical power than the balanced design as $\sigma_2$ deviates more from $\sigma_1$ (the lower right panel of Fig 1).

Departures from the normality assumption were tested under the following distributions: t-(10), t(5), Beta(5, 10), Beta(10, 5), and $\chi^2(3)$. To match means ($\mu_1$ and $\mu_2$) and standard deviations ($\sigma_1$ and $\sigma_2$) of each scenario, these distributions were standardized, scaled by the standard deviations, then shifted by the means. The overall patterns, superiority of the adaptive design over the balanced design, were similar to Fig 1. The adaptive design showed a higher Type I error rate (about 0.08) than the balanced design when data were generated under the chi-square distribution scaled by $\sigma_1 = \sigma_2 = 10$. The balanced design showed a higher Type I error rate (about 0.07) when data were generated under the chi-square distribution $\sigma_1 = 10$ and $\sigma_2 = 100$.

## 2.3. Note

An adaptive design would be more applicable in pilot studies which require relatively short time. A large branch in agricultural sciences relies on field experiments with annual crops, so there may be a practical challenge to use an adaptive design. On the other hand, conclusions of scientific studies are more convincing when data show a consistent pattern between two seasons. Some journals, reviewers, and researchers prefer to see consistent results by repeating an experiment. In addition, count data (e.g., weed count) often violate the normality assumption and homogeneity assumption. A large sample size often mitigates the violation of normality assumption but not the violation of homogeneity assumption. If a large sample size is available, there is no reason to make the homogeneity assumption in the two-sample t-test. In this case, an adaptive design (if it is feasible) provides experimenters an opportunity to increase statistical power by considering an optimal distribution of experimental units between groups. In this process, the unequal variances can be estimated after the first phase of an experiment in order to plan the second phase.

There may be other kinds of weed control studies and related studies. In particular, count data naturally involve non-normality and homogeneity (i.e., data do not follow a normal distribution with equal variance), and there are generalized linear models which can properly account for the uncertainty associated with count data [37–39].

## 3. Comparing time to weed emergence

Traditionally, treatments for weed control have been compared by the average count of weeds per given area [2, 7, 8], the average biomass of weeds per given area [2, 7], and the proportion of area covered by weed colors [9, 10]. The response variables have been recorded at an arbitrary time point. In a cross-sectional assessment, the quantification of effect size may heavily depend on the time of assessment. For example, as shown in Fig 2, the effect of an active

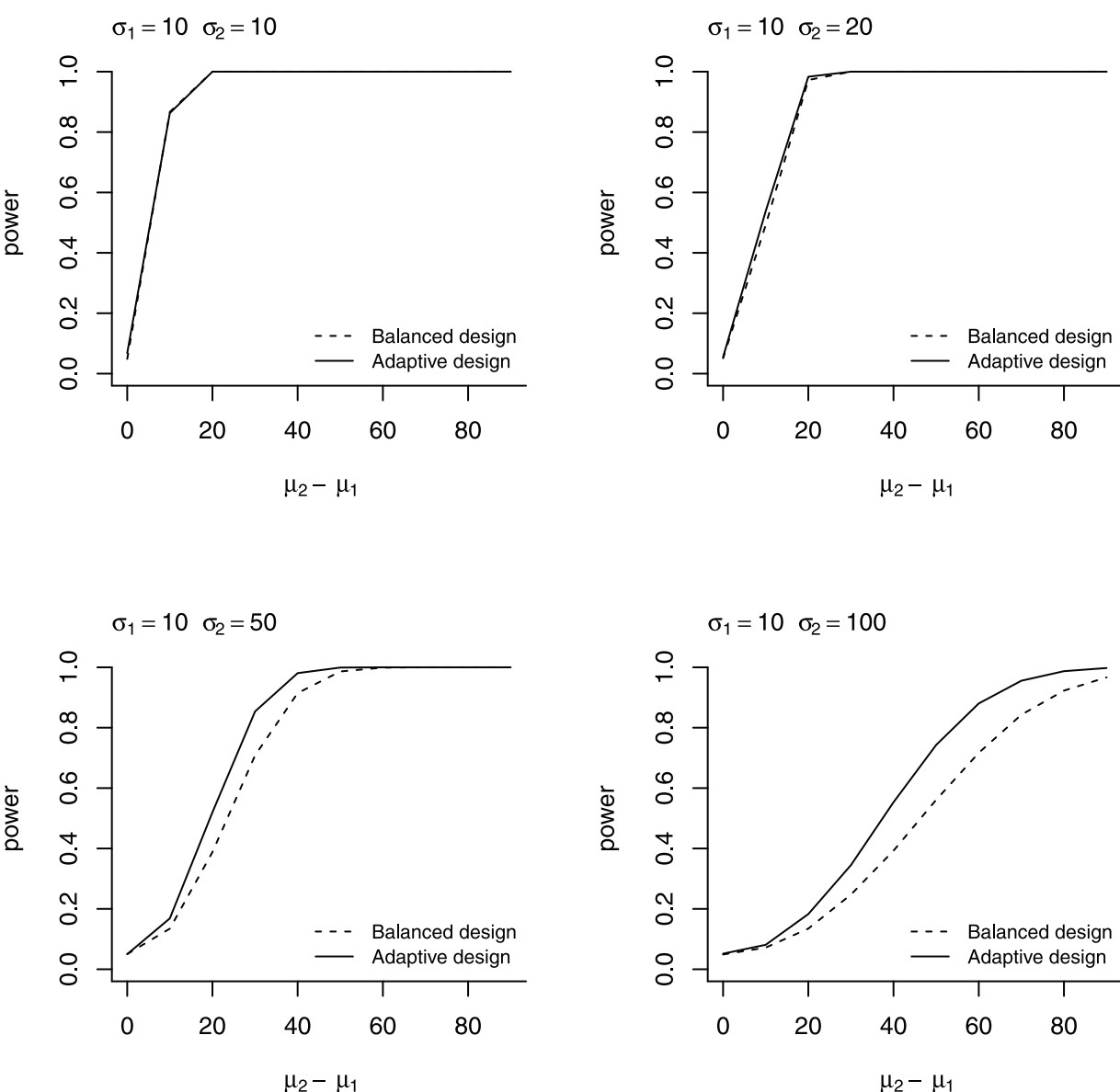

**Fig 1. Power analysis.** This figure compares statistical power between the balanced design and the adaptive design with respect to $\mu_2 - \mu_1$ at $\sigma_1 = 10$ and $\sigma_2 = 10, 20, 50, 100$.

treatment may be similar to the control at the beginning of an experiment, the relative effect size becomes large for a period of time, then the relative effect size eventually becomes the same as the control because, even where pesticides and fumigants have been treated, weeds may eventually emerge.

From farmers' perspective, the primary interest would be how long a treatment delays the weed emergence relative to control. In addition, if a treatment is known to be effective, the question of interest would be how strong (concentration or frequency of an active treatment) the treatment should be in order to balance among cost, effect, and other practical considerations. In this section, we discuss an experimental design to estimate a parameter which quantifies the treatment effect in terms of the time to weed emergence.

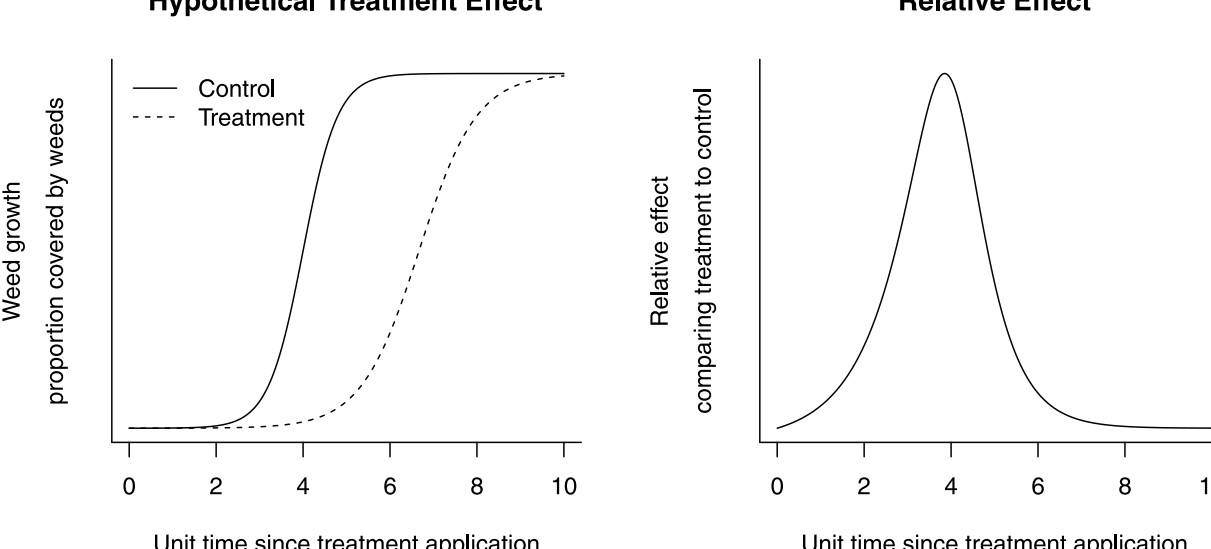

**Fig 2. Treatment effect with respect to time since treatment application.** This hypothetical scenario compares the weed density (proportion of area covered by weeds) between the treatment group and the control group with respect to time since treatment application. The quantification of treatment effect (relative to control) may be highly sensitive to the time of data collection.

## 3.1. Model assumptions

Let $T$ be the waiting time (days) to observe weed emergence, and let $x$ be the ethanol concentration (fixed by an researcher), where $x = 0$ denotes the concentration of 0% (control) and $x = 1$ denotes the ethanol concentration of 100%. We assume $\ln(T) \sim N(\mu_x, \sigma)$, where $\mu_x = \beta_0 + \beta_1 x + \beta_2 x^2$ with the parameter space $-\infty < \beta_0 < \infty, \beta_1 > 0, \beta_2 > 0$, and $\sigma > 0$. Two inequalities $\beta_1 > 0$ and $\beta_2 > 0$ imply that $\mu_x$ increases with respect to $x$ for $0 < x < 1$, and these assumptions will simplify some mathematical subtlety. Let $\Delta$ be the concentration such that $\mu_\Delta = \mu_0 + \ln(2)$ as demonstrated in Fig 3.

The choice of log-normal distribution allows the following interpretation. Under the model assumption, the median of $\ln(T)$, denoted by $\mathcal{M}[\ln(T)]$, and the expectation of $\ln(T)$, denoted by $\mathcal{E}[\ln(T)]$, are equal. Therefore,

$$
\begin{aligned}
\mathcal{M}[\ln(T) \mid x = \Delta] - \mathcal{M}[\ln(T) \mid x = 0] &= \mathcal{E}[\ln(T) \mid x = \Delta] - \mathcal{E}[\ln(T) \mid x = 0] \\
&= \mu_0 + \ln(2) - \mu_0 \\
&= \ln(2).
\end{aligned}
\tag{1}
$$

Further note that

$$
\begin{aligned}
\mathcal{M}[\ln(T) \mid x = \Delta] - \mathcal{M}[\ln(T) \mid x = 0] &= \ln[\mathcal{M}(T \mid x = \Delta)] - \ln[\mathcal{M}(T \mid x = 0)] \\
&= \ln\left[\frac{\mathcal{M}(T \mid x = \Delta)}{\mathcal{M}(T \mid x = 0)}\right].
\end{aligned}
\tag{2}
$$

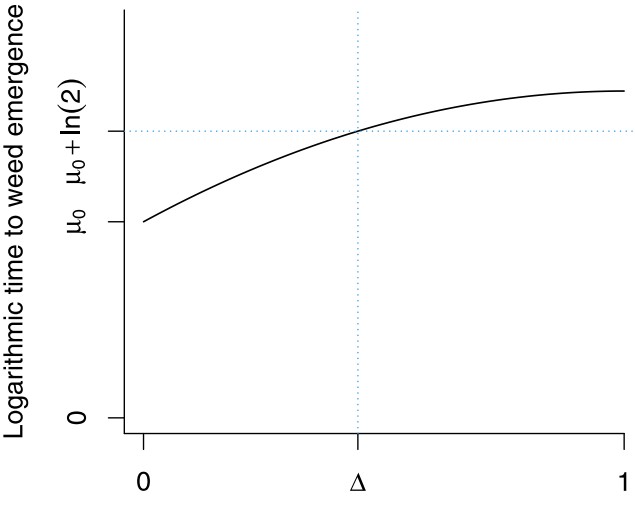

**Fig 3. The relation between Δ and $\mu_\Delta$.** The y-axis represents the logarithmic time to weed emergence, and the x-axis represents the ethanol concentration. The value of Δ corresponds to the concentration such that the expected time to weed emergence increases by ln(2). In other words, the median time doubles at the concentration of Δ when compared to the zero (control) concentration.

From Eqs (1) and (2), we obtain

$$\frac{\mathcal{M}(T \mid x = \Delta)}{\mathcal{M}(T \mid x = 0)} = 2 \,.$$

In the subsequent applied example (Section 4), the primary parameter of interest is Δ, the concentration that corresponds to the doubled median waiting time when compared to the control. Note that

$$\mu_\Delta - \mu_0 = \beta_1 \Delta + \beta_2 \Delta^2 = \ln(2)$$

is equivalent to the quadratic equation

$$\beta_2 \Delta^2 + \beta_1 \Delta - \ln(2) = 0 \,.$$

Under the model assumptions, by the quadratic formula, the parameter Δ has the closed-form expression

$$\Delta = \frac{-\beta_1 + \sqrt{(\beta_1)^2 + 4\ln(2)\beta_2}}{2\beta_2} \,. \tag{3}$$

Note that the choice of the constant 2 (doubled median waiting time) is arbitrary. For any constant $k > 1$ (an increase in median waiting time by $k$ times), ln(2) in Eq (3) can be replaced by k.

### 3.2. Experimental design

Let $x_i$ be a fixed concentration and let $m_i$ be the number of units allocated to $x_i$ for $i = 1, \ldots, k$. Suppose $n$ is the total sample size which is available for an experimenter, so the experimenter

designs an experiment by choosing $(m_1, \ldots, m_k)$ such that $m_1 + \cdots + m_k = n$. Let $T_{ij}$ be the waiting time to be observed at concentration $x_i$ for $j = 1, \ldots, m_i$.

Assuming $\ln(T_{ij}) \sim N(\mu_{x_i}, \sigma^2)$, where $\mu_{x_i} = \beta_0 + \beta_1 x_i + \beta_2 (x_i)^2$ is the assumed quadratic regression, let $\phi_{ij}$ be the normal probability density function for $\ln(T_{ij})$. Given the model parameters $\vec{\theta} = (\beta_0, \beta_1, \beta_2, \sigma^2)$, under the independence assumption, the likelihood function is given by $\mathcal{L}(\vec{\theta}) = \prod_{i=1}^{k} \prod_{j=1}^{m_i} \phi_{ij}$. Given the likelihood function, the Fisher information is defined as

$$\mathcal{I}(\vec{\theta}) = -E\left(\frac{\partial^2 \ln[\mathcal{L}(\vec{\theta})]}{\partial \vec{\theta} \, \partial \vec{\theta}^T}\right),$$

and $\mathcal{V}(\vec{\theta}) = \nabla \vec{h}^T(\vec{\theta}) \, [\mathcal{I}(\vec{\theta})]^{-1} \, \vec{h}(\vec{\theta})$ is the approximation for the variance of the maximum likelihood estimator for $\Delta$ in Eq (3), where

$$\vec{h}^T(\vec{\theta}) = \begin{pmatrix} \frac{\partial \Delta}{\partial \beta_0} & \frac{\partial \Delta}{\partial \beta_1} & \frac{\partial \Delta}{\partial \beta_2} & \frac{\partial \Delta}{\partial \sigma^2} \end{pmatrix}$$

$$\frac{\partial \Delta}{\partial \beta_0} = 0$$

$$\frac{\partial \Delta}{\partial \beta_1} = -\frac{\Delta}{\sqrt{(\beta_1)^2 + 4\ln(2)\beta_2}}$$

$$\frac{\partial \Delta}{\partial \beta_2} = -\frac{1}{\beta_2}\left(\Delta - \frac{\ln(2)}{\sqrt{(\beta_1)^2 + 4\ln(2)\beta_2}}\right)$$

$$\frac{\partial \Delta}{\partial \sigma^2} = 0.$$

This experimental design is referred to as the c-optimal design, and it has been introduced and applied in other regression models [34, 40–42]. The c-optimal design is devised to minimize the expected asymptotic variance of the maximum likelihood estimator for the parameter of interest [40, 43]. The primary focus is to increase the precision in the estimation for the parameter $\Delta$ by seeking the distribution of $(m_1, \ldots, m_k)$ which minimizes the expected value of $\mathcal{V}(\vec{\theta})$ given prior knowledge modeled by a prior distribution $f(\vec{\theta})$. In other words, the c-optimal design minimizes $\int \mathcal{V}(\vec{\theta}) f(\vec{\theta}) \, d\vec{\theta}$ with respect to $(m_1, \ldots, m_k)$.

In agricultural studies, the balanced design (equal replication per group) seems common, and the c-optimal design and other designs often outperform the balanced design for parameter estimation. When researchers have a specific parameter to be estimated, the c-optimal design is devised for the purpose. For a situation when there are multiple criteria to be optimized, robust designs have been discussed [40].

### 3.3. Simulations

To demonstrate the performance of the c-optimal design relative to the balanced design (i.e., $m_i = n/k$ for $i = 1, 2, \ldots, k$), four simulation scenarios were designed as shown in Fig 4. In the figure, the curves represent the expected time to weed emergence in the original unit (days) under the assumption of $\sigma = 1$. For each scenario, $k = 5$ concentrations were fixed at $x_1 = 0$, $x_2 = 1/8$, $x_3 = 1/4$, $x_4 = 1/2$, and $x_5 = 1$, and the total sample size was fixed at $n = 100$. We

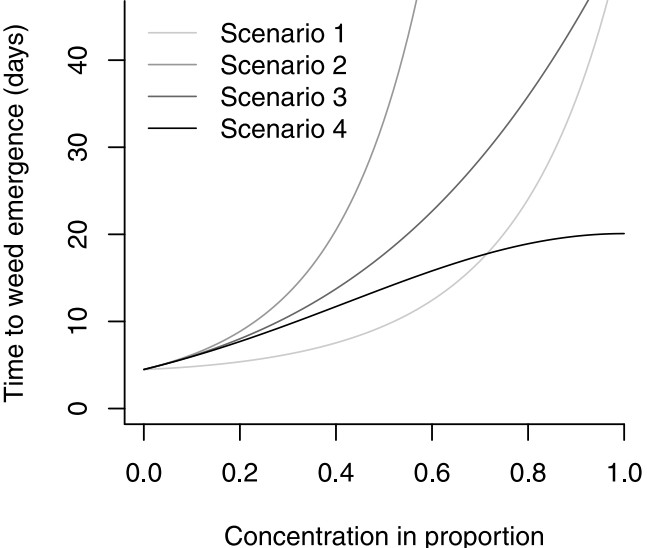

**Fig 4. Simulation scenarios.** The curves are designed by the values of regression parameters $(\beta_0, \beta_1, \beta_2)$ given in Table 1. The true values of $\Delta$ (the parameter to be estimated) are 0.477, 0.203, 0.241, and 0.267 in Scenarios 1, 2, 3, and 4, respectively.

compared the balanced design $\vec{m} = (20, 20, 20, 20, 20)$ and the c-optimal design with a flat prior $f(\vec{\theta}) \propto 1$ which allocated $\vec{m} = (42, 0, 0, 50, 8)$ at the fixed concentrations, respectively.

Each scenario was repeated 1,000 times. All computational work in this simulation and subsequent simulations was done by writing codes with built-in functions in R Version 4.0.2 [36]. The bias, variance, and mean square error (MSE) of posterior mean for $\Delta$ were compared between the two experimental designs as shown in Table 1. It demonstrates the outperformance of the c-optimal design over the balanced design in terms of the three criteria.

## 4. Right-censored time to weed emergence

In the previous section, we considered time to weed emergence as a response variable of interest. In practice, it is implausible to wait for weed emergence in all experimental units because it will require a too long study time. Suppose the maximum time of observation is fixed before initiating an experiment. For instance, in the small-scale experiment to be introduced in Section 4.3, we fixed the maximum time of observation at 30 days, and weed did not emerge until 30 days in some experimental units. In this case, we do not know the exact time of weed emergence, but we know that it is at least 30 days. This type of data is referred as right-censored data, and we revisit the regression model discussed in Section 3 to account for the right-censored data.

**Table 1. Simulation results.**

| Scenario | True parameter values | | | | Balanced design | | | c-optimal design | | |
|---|---|---|---|---|---|---|---|---|---|---|
| | $\beta_0$ | $\beta_1$ | $\beta_2$ | $\Delta$ | Bias | Variance | MSE | Bias | Variance | MSE |
| 1 | 1.0 | 0.5 | 2.0 | 0.477 | + 0.003 | 0.104 | 0.104 | + 0.001 | 0.078 | 0.078 |
| 2 | 1.0 | 3.0 | 2.0 | 0.203 | + 0.012 | 0.052 | 0.053 | + 0.006 | 0.035 | 0.035 |
| 3 | 1.0 | 3.0 | -0.5 | 0.241 | + 0.029 | 0.082 | 0.087 | + 0.010 | 0.054 | 0.055 |
| 4 | 1.0 | 3.0 | -1.5 | 0.267 | + 0.056 | 0.127 | 0.139 | + 0.025 | 0.080 | 0.084 |

## 4.1. Model assumptions

We maintain all assumptions made in Section 3.1 and introduce the following notations. Let $T_{ij}$ denote the actual time of weed emergence in the $j$-th experimental unit of the $i$-th concentration level for $i = 1, \ldots, k$ and $j = 1, \ldots, m_i$. Let $C_{ij} = 1$ if $T_{ij} \leq 30$ (so that the actual time of weed emergence is observed) and $C_{ij} = 0$ if $t_{ij} > 30$ (the actual time is not observed). The likelihood function given in Section 3.1 is modified as $\mathcal{L}(\vec{\theta}) = \prod_{i=1}^{k} \prod_{j=1}^{m_i} (\phi_i)^{c_{ij}} (1 - \Phi)^{1-c_{ij}}$, where $\phi_i$ is the probability density function of $\ln(T_{ij}) \sim N(\mu_{x_i}, \sigma^2)$ and $\Phi_i$ is its cumulative distribution function [44, 45].

## 4.2. Prior specification

Instead of the flat prior $f(\vec{\theta}) \propto 1$, we modeled an informative prior before starting the experiment (to be introduced in Section 4.3). Under the regression model $\mu_{x_i} = \beta_0 + \beta_1 x_i + \beta_2 (x_i)^2$, $\beta_0$ is interpreted as $\mathcal{E}[\ln(T) \mid x = 0]$, and it is equivalent to $\ln[\mathcal{M}(T \mid x = 0)]$ because of the log-normal assumption. Instead of a prior specification on $\beta_0$, we specified a prior on $e^{\beta_0} = \mathcal{M}(T \mid x = 0)$, the median time to weed emergence at the control dose. (This parameterization was easier to elicit our prior knowledge.) We assumed that the median time at the control is shorter than 7 days with a probability 0.5, $P(e^{\beta_0} < 7) = P(\beta_0 < \ln(7)) = 0.5$. We were fairly certain that the median time is shorter than 30 days, and we chose $P(e^{\beta_0} < 30) = P(\beta_0 < \ln(30)) = 0.975$ for computational simplicity. Using a normal prior $\beta_0 \sim N(a_0, b_0)$, we calculated $a_0 = \ln(7) = 1.95$ and $b_0 = 0.5 \ln(30/7) = 0.73$ to reflect the prior assumptions on the median time $e^{\beta_0}$.

Under the regression model, it was challenging to elicit a prior distribution jointly on $\beta_1 > 0$ and $\beta_2 > 0$ in a tractable way. For the sake of simplicty, we specified $\beta_1 \sim \text{Exp}(d_1)$ and $\beta_2 \sim \text{Exp}(d_2)$ independently. The hyper-parameters, $d_1$ and $d_2$, were chosen by trial and error such that $P(\Delta < 0.5) \doteq 0.95$ and $P(\Delta < 1) \doteq 1$, where $\Delta$ is the transformation of $\beta_1$ and $\beta_2$ given in Eq (3). After several iterations of trial and error, we found that $d_1 = 0.2$ and $d_2 = 0.2$ are reasonable. For the standard deviation $\sigma > 0$, a flat prior was chosen independently.

## 4.3. Adaptive experiment design (applied example)

Typical weed control treatments contain pre- and/or post-emergence pesticides, fumigants, biofumigants, solarization, flaming, and hand hoeing [1]. While pesticide-based weed controls are known to be biologically efficacious and economically efficient, most of them are harmful to environments. Consumers have raised their concern, they have showed a high interest in organic products, and regulations on the use of pesticides have been strengthened. Ethanol (EtOH) contained in plants or synthesized in factories is an easily available, low-toxic solvent. Although EtOH is not registered as a biological control agent, researchers have reported that it inhibits the germination of weed seeds. For instance, it was shown that the germination of morning glory seeds was reduced after being exposed to 1% v/v of EtOH [46]. Since EtOH is a natural product, EtOH may be available as a biological control agent. It seems promising that a high concentration of EtOH is effective, and our objective is to find an adequate concentration. We acknowledge that there are more realistic powerful herbicides in weed science, and this section is devised for the purpose of demonstrating the adaptive experimental design for estimating the parameter $\Delta$. An experiment of EtOH was conducted to find the concentration which doubles the median time of weed emergence, when compared to the control, and this parameter is denoted by $\Delta$ as in Section 3.

Each flowerpot contained 10 seeds of ryegrass (*Lolium multiflorum*). At the center of each flowerpot, we prepared to apply 15 mL of 0% (non-treated control), 12.5%, 25%, 50%, and 100% of EtOH were applied. That is, we fixed the experimental concentrations at $x_1 = 0$, $x_2 = 0.125$, $x_3 = 0.25$, $x_4 = 0.5$, and $x_5 = 1$. The original experimental plan was to have a sample of size $n = 100$, but only 50 flowerpots were available at a time. We decided to perform an adaptive experimental study, and we fixed the maximum waiting time of 30 days per phase because the emergence of ryegrass would take an extremely long time at a high concentration of EtOH.

For the first phase of this experiment, we applied the c-optimal design using the prior in Section 4.2. The c-optimal design allocated $m_1 = 11$ flowerpots at $x_1 = 0$, $m_4 = 25$ flowerpots at $x_4 = 0.5$, and $m_5 = 14$ flowerpots at $x_5 = 1$. All flowerpots were monitored daily. All of the 11 flowerpots at $x_1 = 0$ had ryegrass emerged within 30 days (average of 12.45 days), 13 out of the 25 flowerpots at $x_4 = 0.5$ had ryegrass emerged within 30 days, and none of the 14 flowerpots at $x_5 = 1$ had ryegrass emerged within 30 days.

After collecting the data in the first phase, we combined the prior $f(\vec{\theta})$ with the likelihood $\mathcal{L}(\vec{\theta})$ for the posterior, and we applied the c-optimal design for the next 50 flowerpots by minimizing the posterior expectation of $V(\vec{\theta})$. Note that we used the likelihood $\mathcal{L}(\vec{\theta})$ of the form given in Section 4.1 to account for the right-censored data. For the second phase, the c-optimal design allocated 32 flowerpots at $x_1 = 0$ and 18 flowerpots at $x_4 = 0.5$. In other words, the c-optimal design suggested stop observing at the maximum (100%) concentration, and it attempted to gather more information at the control than at the 50% concentration in order to reduce uncertainty about $\Delta$. For the observed data, see S1 and S2 Data.

Fig 5 graphically presents the change in the knowledge about $\Delta$ before the experiment (prior) and after the first phase and the second phase of the experiment (posteriors). The respective point estimates for $\Delta$, using the mean of distribution, were 0.21, 0.46, and 0.39, respectively. The respective 95% credible intervals were (0.04, 0.7), (0.35, 0.6), and (0.33, 0.45), and the degree of uncertainty about $\Delta$ decreased as we collected more data.

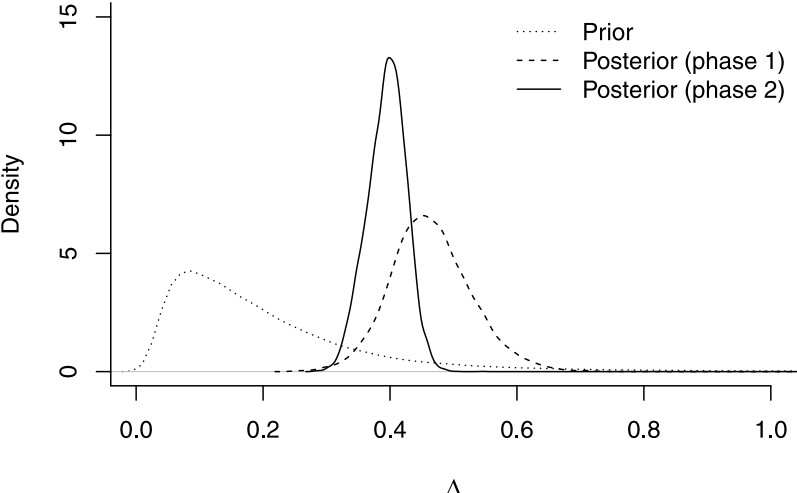

**Fig 5. Prior and posterior inference for $\Delta$.** This figure demonstrates that the uncertainty about $\Delta$ decreases as data are accumulated.

**Table 2. Simulation results.**

| Scenario | True parameter values | | | | Fixed design | | | Adaptive designs | | |
|---|---|---|---|---|---|---|---|---|---|---|
| | $\beta_0$ | $\beta_1$ | $\beta_2$ | $\Delta$ | Bias | Variance | MSE | Bias | Variance | MSE |
| 1 | 2.5 | 0.07 | 1 | 0.798 | -0.064 | 0.148 | 0.161 | -0.054 | 0.150 | 0.160 |
| 2 | 2.5 | 0.07 | 3 | 0.469 | -0.073 | 0.048 | 0.087 | -0.051 | 0.049 | 0.070 |
| 3 | 2.5 | 0.7 | 1 | 0.553 | -0.024 | 0.087 | 0.090 | -0.020 | 0.082 | 0.084 |
| 4 | 2.5 | 0.7 | 3 | 0.378 | -0.037 | 0.046 | 0.059 | -0.025 | 0.042 | 0.049 |
| 5 | 2.5 | 1.7 | 1 | 0.340 | 0.017 | 0.056 | 0.058 | 0.016 | 0.050 | 0.053 |
| 6 | 2.5 | 1.7 | 3 | 0.275 | -0.007 | 0.037 | 0.037 | -0.004 | 0.033 | 0.033 |

### 4.4. Retrospective simulations

After the second phase, the posterior means of $\beta_0$, $\beta_1$, and $\beta_2$ were 2.2, 1.0, and 1.9, respectively. Six simulation scenarios were designed around these parameter values. This was a retrospective simulation study to investigate what would happen if the c-optimal design was performed at once instead of taking two phases. Each retrospective scenario was replicated 1,000 times. The adaptive design resulted in lower MSEs in the six scenarios as shown in Table 2.

### 4.5. Note

In a large-scale field experiment, random-effects may exist due to random germination times and other environmental factors. Additional sources of random variations can be modeled via a mixed-effects model. An experimental design under a mixed-effects model requires a more sophisticated variance structure, the underlying mathematical formulas are much more technical [47–49]. In addition, survival analysis (analyzing time-to-event data) in weed science has been discussed in literature [50–52].

## 5. Synergistic effect

Sometimes researchers seek a synergistic effect of two treatments [53–56]. Suppose an outcome is coded as 1 or 0 (e.g., 1 for suppressing germination, 0 otherwise), and let $\pi$ be the probability of the outcome coded as 1. Let $x$ be the concentration of treatment A and $z$ be the concentration of treatment B. The logistic regression is given by

$$\pi_{(x,y)} = \frac{e^{\beta_0+\beta_1 x+\beta_2 z+\beta_3 xz}}{1+e^{\beta_0+\beta_1 x+\beta_2 z+\beta_3 xz}},$$

or equivalently

$$\ln\left(\frac{\pi_{(x,y)}}{1-\pi_{(x,y)}}\right) = \beta_0 + \beta_1 x + \beta_2 z + \beta_3 xz.$$

Under the model, the parameter of interest is $\beta_3$ to test for the presence of synergistic or antagonistic effect between the two treatments. If $\beta_3 = 0$, the null hypothesis, it implies the absence of synergistic or antagonistic effect. If $\beta_3 \neq 0$, the alternative hypothesis, it implies the presence of synergistic or antagonistic effect.

For the purpose of demonstration, suppose a researcher has four concentrations for treatment A, say $x = 0, 0.25, 0.5, 1$, and four concentrations for treatment B, say $z = 0, 0.25, 0.5, 1$. If the researcher can afford a total sample of size 160, there are 10 units allocated to each possible combination $(x, y)$ for a balanced design. Instead of the balanced design, the researcher may consider the d-optimal design which maximizes the determinant of the Fisher expected

information (FEI) matrix, and it is devised to increase the amount of information about the model parameters $\vec{\beta} = (\beta_0, \beta_1, \beta_2, \beta_3)$ globally. Alternatively, the c-optimal design maximizes the asymptotic variance of the maximum likelihood estimator for $\beta_3$ (i.e., the (4,4)-th element of the inverted FEI), and it is devised to maximize the information about the target parameter $\beta_3$ in order to test for the synergistic or antagonistic effect.

### 5.1. Prior specifications for simulation study

For the d- and c-optimal designs, we need a prior specification on $\vec{\beta}$. Agriculture researchers may collaborate with statisticians to express a prior (researchers' knowledge prior to an experiment) via a probability distribution. Instead of directly expressing a prior on $\vec{\beta}$, which is difficult to interpret in the context of research, prior knowledge can be expressed on the probability of an outcome at four (the number of regression parameters) arbitrary concentrations. For the purpose of demonstration, we considered four concentrations (0, 0), (1, 0), (0, 1), and (1, 1), and we specified $\pi_{(0,0)} \sim \text{Beta}(1, 1)$, $\pi_{(1,0)} \sim \text{Beta}(1, 1)$, $\pi_{(0,1)} \sim \text{Beta}(1, 1)$, and $\pi_{(1,1)} \sim$ Beta(1, 1) independently to express a high degree of uncertainty. This non-informative prior is referred to as prior 1 in this section. The independent priors on $\pi$'s can be transformed to the joint prior of $\vec{\beta} = (\beta_0, \beta_1, \beta_2, \beta_3)$ as shown in the left panel of Fig 6. This method of eliciting a prior distribution on $\vec{\beta}$ is known as the conditional mean prior [57]. To express a less degree of uncertainty, we specified $\pi_{(0,0)} \sim \text{Beta}(2, 8)$, $\pi_{(1,0)} \sim \text{Beta}(5, 5)$, $\pi_{(0,1)} \sim \text{Beta}(5, 5)$, and $\pi_{(1,1)} \sim$ Beta(8, 2) independently. This prior is referred to as prior 2, and the informative prior leads to smaller prior variances on $\vec{\beta}$ as shown in the right panel of Fig 6.

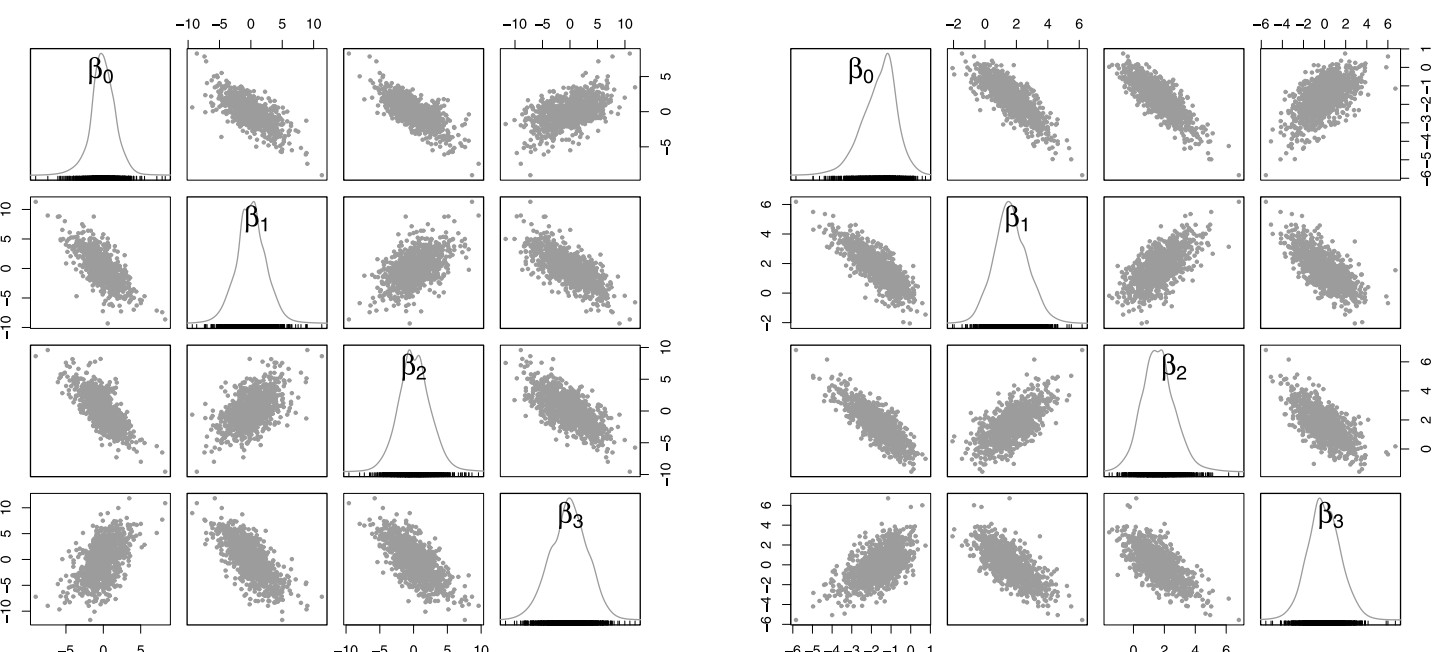

**Fig 6. Joint prior distributions of model parameters.** The scatter plots graphically demonstrate the joint prior distribution of $\vec{\beta} = (\beta_0, \beta_1, \beta_2, \beta_3)$ induced from the independent beta priors on $\pi_{(0,0)}$, $\pi_{(1,0)}$, $\pi_{(0,1)}$, and $\pi_{(1,1)}$. The figure on the left demonstrates prior 1 (non-informative prior), and the figure on the right demonstrates prior 2 (informative prior).

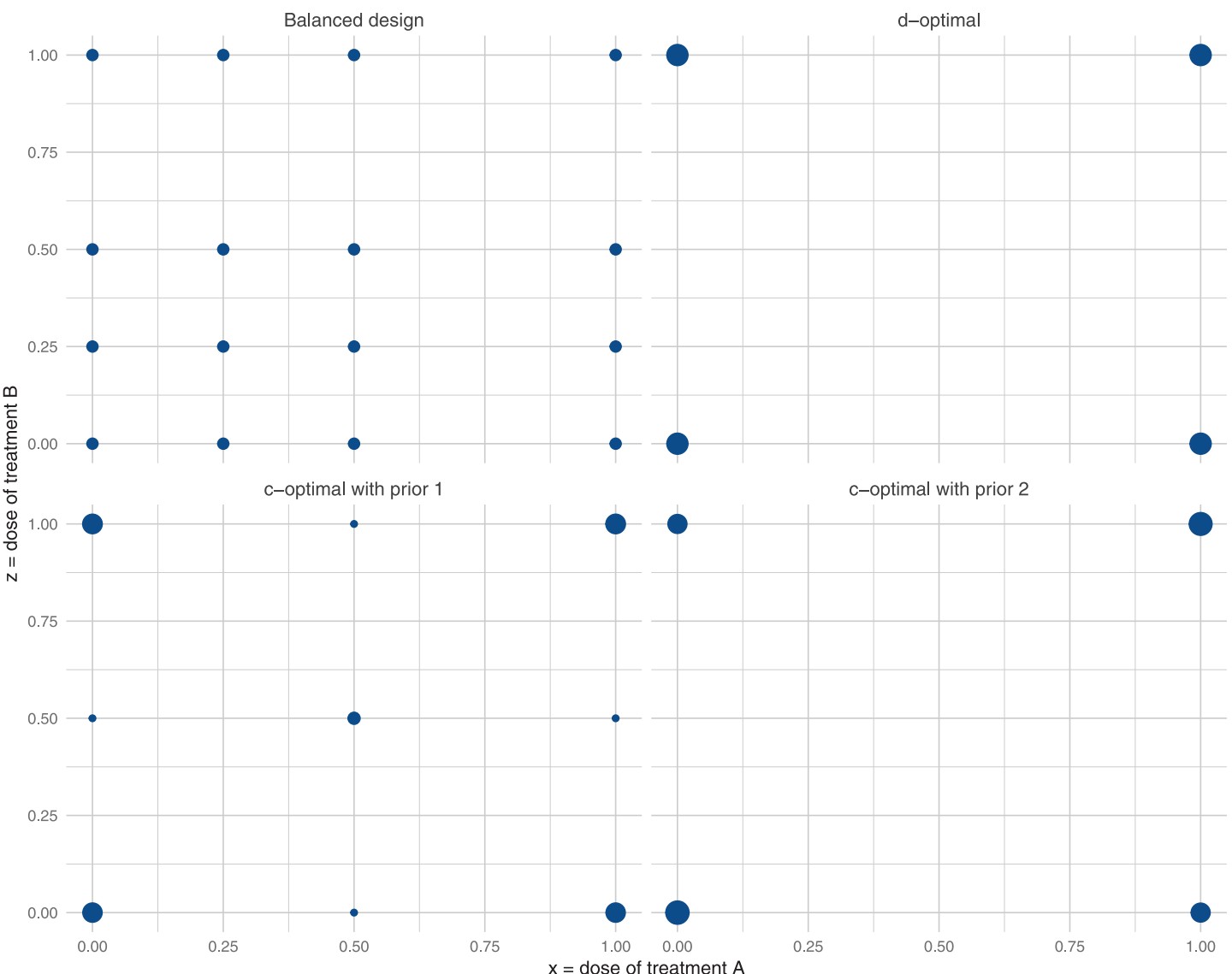

**Fig 7. Experimental designs.** Unlike the balanced design (the same proportion of experimental units across all concentration points), the d-optimal design and the c-optimal design with prior 2 distribute experimental units at the extreme points (0, 0), (0, 1), (1, 0), and (1, 1). The c-optimal design with prior 1 (non-informative prior) seeks information at a variety of concentration points.

## 5.2. Experimental designs

The c-optimal design is sensitive to a prior specification, but the d-optimal design is not. In Fig 7, the four experimental designs (balanced, d-optimal, c-optimal with prior 1, and c-optimal with prior 2) are compared in terms of the relative proportion of units (out of the total 160) allocated at the 16 possible combinations of two treatments. The d-optimal design spreads the total sample size of 160 evenly at the four concentration points $(x, z) = (0, 0), (0, 1), (1, 0),$ and $(1, 1)$. In other words, it widely spreads the units on the entire concentration space $[0, 1] \times [0, 1]$ in order to learn about all model parameters $\vec{\beta} = (\beta_0, \beta_1, \beta_2, \beta_3)$ globally. The c-optimal design with prior 1 balances between the extreme and middle concentrations at some degree, but the c-optimal design with prior 2 resembles the d-optimal design.

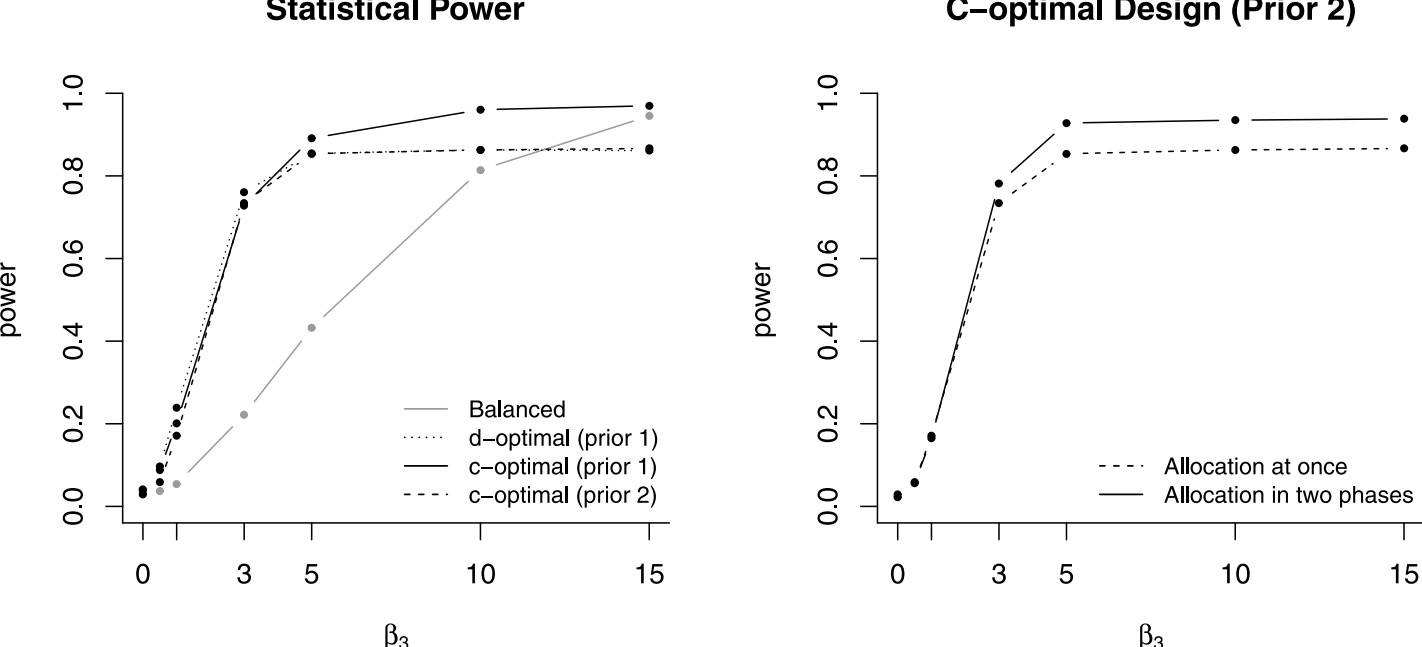

**Fig 8. Statistical power.** The figure on the left demonstrates that the c-optimal design and d-optimal design outperform the balanced design when $\beta_3$ is relatively close to the null value 0. The figure on the right demonstrates that the c-optimal design with prior 2 can be improved by the adaptive design.

### 5.3. Simulations

To compare the four designs, seven simulation scenarios were considered. The values of $\vec{\beta}$ were chosen at $\beta_0 = -1.5$, $\beta_1 = 0.75$, $\beta_2 = 1.5$, and $\beta_3 = 0, 0.5, 1, 3, 5, 10, 15$ to vary the degree of synergistic effect. Each scenario was replicated 1,000 times to approximate statistical power for testing $\beta_3 = 0$ versus $\beta_3 \neq 0$ at significance level 0.05 for each design. As shown in the left panel of Fig 8, the d-optimal design outperforms the balanced design until $\beta_3 = 10$, and the c-optimal design with prior 1 outperforms the d-optimal design. The power of c-optimal design with prior 2 seems similar to the power of d-optimal design because the two designs were very similar as shown in Fig 7. When $\beta_3 = 15$ the balanced design outperformed the d-optimal design and the c-optimal design with prior 2.

The power of c-optimal with prior 2 was substantially lower than the power with prior 1 for high values of $\beta_3$, and it is because the strong prior substantially deviated from the true simulation scenarios. The power could be improved by implementing adaptive design. In the first phase, 80 units were allocated based on prior 2, and the remaining 80 units were allocated based on posterior (prior 2 and collected data). The two-step procedure was helpful to correct the initial c-optimal design, and the power was noticeably increased for $\beta_3 \geq 3$ as shown in the right panel of Fig 8.

### 5.4. Note

Binomial counts are typically over-dispersed which means the data are more variable than the assumption under the standard logistic regression discussed in this section. The over-dispersion can be addressed via a mixed-effects model or a quasi-binomial logistic regression. The quasi-binomial model includes a dispersion parameter, and it scales the standard error of under the standard logistic regression. Regardless, the c-optimal design has the same objective which is to reduce uncertainty associated with the estimation for the parameter of interest.

## 6. Discussion

A clear objective of an experiment should be specified before choosing an appropriate experimental design [58]. This point was demonstrated in Section 5.3. If an objective was to investigate the interaction between two treatments, the c-optimal design would result in higher statistical power than the d-optimal design. Sometimes a researcher has multiple objectives, and this situation has been discussed in the context of a non-monotonic dose-response relationship in toxicology [40]. Choosing an objective-specific experimental design is not a new idea. It has been practiced among engineers and drug developers [59]. Like other research areas, agricultural data are expensive in terms of time and effort given a fixed amount of resources. Therefore, a careful experimental design is worth to be considered before initiating an experiment.

Hopefully this article alleviates some misconceptions of balanced designs. It is an optimal approach under specific cases like when two groups have the same variance in the two-sample t-test, but $\sigma_1 = \sigma_2$ or $\sigma_1 \neq \sigma_2$ is out of researcher's control. After a researcher gains information about $\sigma_1$ and $\sigma_2$, via a pilot study or the first phase of a multi-phase experiment, the researcher may attempt to balance between $\sigma_1^2/n_1$ and $\sigma_2^2/n_2$ by choosing appropriate $n_1$ and $n_2$. In practice, a researcher may face a situation when some treatments might be more difficult to run or more expensive than other treatments [60]. Therefore, an experimental design is a practical problem of balancing between statistics and logistics.

In this article, given a specific objective which is formulated by a model parameter, we discussed adaptive designs to address uncertainty about researcher's prior knowledge. Scientific research requires some degree of assumptions prior to data collection, and an adaptive design provides an opportunity to correct the prior assumption before exhausting all available resources. If the initial assumption is reasonably close to the truth, an adaptive design will not be detrimental as shown in the simulations of this article. Despite an adaptive design being a practical challenge because the total time of an experiment would be increased, we believe that the benefit of an adaptive design is clear from statistical perspective.

In agricultural studies, it is common to collect data for two seasons to confirm a hypothesis [1, 2, 8, 61]. It is also an opportunity to consider an adaptive design or some variation as there is no single statistical strategy which can fit all situations. Collaborations between agricultural researchers and statisticians are highly encouraged to find an appropriate strategy for a given research objective under practical and logistical considerations. Simulating data and comparing multiple possible plans under likely scenarios would be a recommended practice.

## 7. Conclusion

A research question can be formulated via a statistical parameter (a quantity which measures the treatment effect), and an experiment can be designed to increase the amount of information about the parameter of interest. In practice, increasing the sample size is not always feasible, so researchers fix a sample size at their maximal capacities. The simulations demonstrated that unbalanced and adaptive designs provide smaller error in parameter estimation and higher statistical power in hypothesis testing than balanced and fixed designs. Therefore, researchers facing different practical situations can utilize available resources efficiently by using appropriate experimental designs.

## Supporting information

**S1 Data. Data observed after the first phase of the experiment.**
(CSV)

**S2 Data. Data observed after the second phase of the experiment (combined with the first phase).**
(CSV)

## Author Contributions

**Conceptualization:** Steven B. Kim, Dong Sub Kim, Christina Magana-Ramirez.

**Data curation:** Steven B. Kim, Christina Magana-Ramirez.

**Formal analysis:** Steven B. Kim, Christina Magana-Ramirez.

**Investigation:** Steven B. Kim, Dong Sub Kim, Christina Magana-Ramirez.

**Methodology:** Steven B. Kim.

**Project administration:** Dong Sub Kim.

**Resources:** Dong Sub Kim.

**Software:** Steven B. Kim.

**Supervision:** Steven B. Kim, Dong Sub Kim.

**Validation:** Steven B. Kim, Dong Sub Kim.

**Visualization:** Steven B. Kim, Christina Magana-Ramirez.

**Writing – original draft:** Steven B. Kim, Dong Sub Kim, Christina Magana-Ramirez.

**Writing – review & editing:** Steven B. Kim, Dong Sub Kim.

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
