## [Decision Letter · Decision Letter 0]

21 Jul 2021

PONE-D-21-17452

Applications of statistical experimental designs to improve statistical inference in weed management

PLOS ONE

Dear Dr. Kim,

Thank you for submitting your manuscript to PLOS ONE. After careful consideration, we feel that it has merit but does not fully meet PLOS ONE’s publication criteria as it currently stands. Therefore, we invite you to submit a revised version of the manuscript that addresses the points raised during the review process.

Please take the consideration every single point that three referees pointed out. 

We look forward to receiving your revised manuscript.

Kind regards,

Ahmet Uludag, Ph.D.

Academic Editor

PLOS ONE

Journal Requirements:

“The authors declare that there is no funding associated with this manuscript.”

Additional Editor Comments:

I have review of three referees. One rejected and the other two concluded minor revision. I think this is an important paper but needs to be improved. Please follow suggestions of three reviewers. There are good suggestions from referee2 as well. DO your best to meet his suggestions where appropriate. I think he gave good insights to add your paper.

Reviewers' comments:

Reviewer's Responses to Questions

**Comments to the Author**

1. Is the manuscript technically sound, and do the data support the conclusions?

Reviewer #1: Yes

Reviewer #2: Partly

Reviewer #3: Yes

2. Has the statistical analysis been performed appropriately and rigorously? 

Reviewer #1: Yes

Reviewer #2: Yes

Reviewer #3: Yes

3. Have the authors made all data underlying the findings in their manuscript fully available?

Reviewer #1: Yes

Reviewer #2: Yes

Reviewer #3: Yes

4. Is the manuscript presented in an intelligible fashion and written in standard English?

Reviewer #1: Yes

Reviewer #2: No

Reviewer #3: Yes

5. Review Comments to the Author

Reviewer #1: The manuscript aims to introduce and explain methodology for better planning of agricultural experiments through the use of optimal designs. This is a good idea as optimal designs aren't used much, also not in agricultural sciences. However, the manuscript could provide even more explanations and insights if it's the intention to have more practitioner use these methods.

Specifically, the manuscript looks at 3 practical situations, which occur frequently in practice. In each situation, theory is introduced very briefly, with some concepts left unexplained, followed by simulation results and possibly a practical example. For instance, no literature on mixture models or optimal design for mixture models are cited; this is disappointing.

Specific comments:

lines 26-27: Are estimation of standard errors really modified?!

lines 59-64: It would be helpful if it was spelled out that three "practical situations" are dealt with in detail in the manuscript.

lines 98-107: Which statistical software was used?

line 160: No reference for the c-optimal design.

lines 244-310: No references to existing literature on mixture models or optimal design for mixture are provided. One place to start:

https://doi.org/10.1002/env.666

https://doi.org/10.1016/j.csda.2017.08.006

Reviewer #2: This paper considers several examples, modeled on applications in weed science, where a good design involves unequal replication. This finding may come as a surprise to some weed scientists, because it is very common in practive to use equally replicated designs.

While the intention of this paper is very laudable, I find most of the examples unconvincing because they are rather artificial. Perhaps the most important limitation of the proposed methods is the need to run an experiment twice, using the first to optimize the design for the second. Most experimenters in weed science will simply not have the time to run two experiments in order to get a single experiment's worth of results. The paper lacks convincing examples, where such designs would really be practical and where weed scientists would indeed be willing to use them.

Major comments

p.1: The Duncan test does not control the family-wise error rate and should not be recommended. The t-test does not either, but the Tukey test does. I think it is important to mention these facts when introducing these popular tests. The current wording "seem to be alternative choices among researchers" leaves the authors' view open at this point and may even raise the impression that the choice does not matter.

p.2: To introduce the idea of an unbalanced design, the authors consider the case of a linear regression with four equally spaced doses. The fact that the best design allocates half the observations to the smallest and largest dose and none to the intermediate ones does not mean the design is balanced. It's quite balanced between the highest and lowest dose. It is a well known fact that with linear regression, only two doses are needed to estimate the regression coefficient and that this is the optimal design. This design does rest on the strong assumption that there is no departure from linearity, and this is a very strong assumption that many practitioners may not be willing to make. This, I would hardly ever recommend such a design to a weed scientist in practice.

p.2, bottom: The authors use the term "two-phase design" for an approach that takes a first example to inform about the parameters of the model to be fitted and then in a second phase adapts the sample size to achieve a desire precision. I know at this point what the authors are intending but notice at the same time that the term "two-phase design" can have quite different meaning in other contexts. See, e.g.,

Brien, C. J. (2019). Multiphase experiments with at least one later laboratory phase. II. Nonorthogonal designs. Australian & New Zealand Journal of Statistics, 61:234-268.

I am more familiar with the term "adaptive design" for the approach the authors are pursing. These types of design are very common in clinical trials, and there is a very rich literature on this kind of design. The authors do not seem to be aware of this literature. Perhaps they can consider adopting this term to avoid confusion. Moreover, delving into the relevant literature may be useful. The authors do mention some work along these lines in a cursory manner at the very end of the paper, but it would be appropriate to state this from the very start, making it clear that the paper presents nothing that is in itself new and that much of the relevant theory was developed in the context of clinical trials, where one can find ample guidance for the types of design the authors are advocating.

p.3: Having introduced linear regression as one example of an "unbalanced design" in the introduction, it is a bit surprising that Section 2 focuses on the comparison of two means. The connection with linear regression is not at all clear. Also, in the introduction the authors mention various examples of traits that involve counts and as such do not obey the normality or homogeneity assumption. Readers would expect that methods for count data are considered, e.g. generalized linear models. It's actually not clear after the introduction what will be the overall structure of the paper and how many different cases will be considered.

p.3: Section 3 really lacks an agricultural example to motivate and illustrate the method. There is a large branch in agricultural sciences that relies primarily on field experiments with annual crops. In these settings there is rarely the luxury to use an adaptive design. The authors are still assuming normality here, whereas the introduction motivated the need to assume heterogeneity of variance using the example of count data. As pointed out before, there are tailored methods for count data, which would be prefered in this setting, and based on the introduction, this is what I would have expected in Section 2.

In Section 3, the authors consider a second example, i.e. a dose-response relationship where the interest is in finding the dose (denoted as Delta) that doubles the time to emergence of a weed. The authors suggest what they denote as a c-optimal design, but they give no reference whatsoever for this standard design problem. The assert that the information about Delta can be increased by minimizing the prior expectation of the variance of the parameter vector, but I am not seeing any proof of this claim. Moreover, what we really want is the optimal design for estimating Delta, not a design that "increases the information about Delta" (compared to what baseline design?). Finally, it is not even explained how the "optimal" number of replications are to be determined, let alone the number of x levels and their placement along the x-axis. All of this looks to be a standard design problem in nonlinear regression for which canned solutions are available but the authors make it seem like an entirely level problem. Showing in simulations that the equally replicated designs can be beaten is knocking a strawman. Incidentally, there is quite a lot of literature on time-to-event data as used in weed science, and there are several good R packages for this kind of data. The lognormal is but one distributional model for such data, and I am wondering why the authors picked this particular distrubution. Perhaps because some of the algebra is straightforeard? What if the times-to-event follow a Weibull or Gompertz distribution, for example?

Section 4.1: I have tried hard to understand the rationale of the computations leading to the prior for this example but failed. Section 4.2 only mentions right-censoring in passing, focussing on the derivation of a design under a right-censored model. Such a model was never mentioned in Section 3, so as a reader I am confused here and do not know what is going on. Why censoring all of a sudden? I think this kind of modelling needs a much more thorough introduction for the intended audience, which I think is agronomists and weed scientists. Censoring is a key property of such data, so just mentioning this in passing as if every body knew this does not seem appropriate for the intended audience. Regarding the example, I note that the authors are assuming 10 seeds per pot. Each seed may have a different germination time, and it is most informative to analyse individual plant data. These data are clustered, however, and this would require fitting random pot effects.

Section 5 considers a logistic regression model to assess synergistic effects between two quantitative inputs. The response is binary, and germination of individual seeds is considered as a case in kind. This is just a hypothetical example, no real experiment is considered. The main challange with this kind of data, again, is clustering. Several seeds are usually placed on the same experimental units (pot, petri dish, filter paper), and the number of germinated counts is assessed per experimental unit. These binomial counts are very typically overdispersed, which is the main challenge in the analysis. The whole derivation in this section ignores overdispersion, so I am afraid it is not relevant for most practical purposes.

In summary, my impression is that the authors do not have a very intimate connection with experimenters and real experiments in the weed sciences and that the methods they propose are not very practical. What I find mainly lacking are convincing real examples that fully motivate the proposed methods. Most of the derivations are based on rather hypothetical and artificial settings.

I do aplaud the general intention of the paper, which I would say is focused on the optimal design for regression models in weed science. I believe, however, that this work would benefit greatly from a closer collaboration with weed scientists. If examples can be presented where the proposed designs have really been found to work for weed scientists, this could make a much more convincing case. This may also mean rather different designs presented as desirable in the end.

Reviewer #3: The study is focused on one of the most important issues in weed science, experimental designs. The concept structured was firstly described and given its theoretical background. Although the data presented by author to clarify their ideas was enough, it would be obtained more realistic and powerful from herbicides instead of EtOH. Under different scenarios, the two-phase design may create lower MSE’s than single stage design.

Please check the sentence in line 80-81 “The null hypothesis is H0: µ1- µ 2 = 0, and the alternative hypothesis is H1: µ1- µ 2 = 0”.

Please check the value in line 216 “x3 = 0.5”.

Please add separate captions for each figure.

6. PLOS authors have the option to publish the peer review history of their article (what does this mean?). If published, this will include your full peer review and any attached files.

Reviewer #1: No

Reviewer #2: No

Reviewer #3: **Yes: **Ahmet Tansel Serim

---

## [Author Response · Author response to Decision Letter 0]

26 Jul 2021

Reviewer #1: The manuscript aims to introduce and explain methodology for better planning of agricultural experiments through the use of optimal designs. This is a good idea as optimal designs aren't used much, also not in agricultural sciences. However, the manuscript could provide even more explanations and insights if it's the intention to have more practitioners use these methods.

Specifically, the manuscript looks at 3 practical situations, which occur frequently in practice. In each situation, theory is introduced very briefly, with some concepts left unexplained, followed by simulation results and possibly a practical example. For instance, no literature on mixture models or optimal design for mixture models are cited; this is disappointing.

Response: Thank you for your feedback. We added more theoretical and conceptual explanations, particularly in Section 4. The section is revised substantially to thoroughly explain the concepts. Furthermore, we added references related to mixed-effects models and optimal designs. [e.g., Lines 205-208; 326-329]

Specific comments:

lines 26-27: Are estimation of standard errors really modified?!

Response: The Tukey-Kramer method adjusts the standard error to account for unequal sample sizes when compared to the traditional Tukey’s method (which was devised for equal sample size). We edited the language to avoid confusion. Thank you for your comment. [Lines 30-33]

lines 59-64: It would be helpful if it was spelled out that three "practical situations" are dealt with in detail in the manuscript.

Response: We think that this is a great idea for readers. We spelled out the three practical situations (parameters of interest) and provided associated section numbers. [Lines 81-85]

lines 98-107: Which statistical software was used?

Response: R Version 4.0.2 was used. It is added and cited in the revised manuscript. [Lines 127-128]

line 160: No reference for the c-optimal design.

Response: References for the c-optimal design are added. Thank you. [Lines 205-206]

lines 244-310: No references to existing literature on mixture models or optimal design for mixture are provided. One place to start:

https://doi.org/10.1002/env.666

https://doi.org/10.1016/j.csda.2017.08.006

Response: Thank you for the suggested references. In addition to your suggested references, we added a couple more. [Line 331]

Reviewer #2: This paper considers several examples, modeled on applications in weed science, where a good design involves unequal replication. This finding may come as a surprise to some weed scientists, because it is very common in practice to use equally replicated designs.

While the intention of this paper is very laudable, I find most of the examples unconvincing because they are rather artificial. Perhaps the most important limitation of the proposed methods is the need to run an experiment twice, using the first to optimize the design for the second. Most experimenters in weed science will simply not have the time to run two experiments in order to get a single experiment's worth of results. The paper lacks convincing examples, where such designs would really be practical and where weed scientists would indeed be willing to use them.

Response: Thank you for your careful review. We agree that some researchers cannot afford the time to run two experiments. This article demonstrates the potential benefit of a two-phase (adaptive) design. We (the authors) did not mean to argue that all experiments must be run twice. Via simulations, we demonstrated that a single-phase optimal design outperforms the balanced design which is commonly seen in literature. We recommend that researchers consider an optimal design (or any appropriate variation) based on their clear objective (a parameter of interest) to increase statistical power. Based on our academic and working experiences, some journal reviewers and practitioners believe that the effect of a treatment is more convincing when the experiment was replicated (two seasons of data). The demonstrated two-phase (adaptive) design can be beneficial in this situation. 

Major comments

p.1: The Duncan test does not control the family-wise error rate and should not be recommended. The t-test does not either, but the Tukey test does. I think it is important to mention these facts when introducing these popular tests. The current wording "seem to be alternative choices among researchers" leaves the authors' view open at this point and may even raise the impression that the choice does not matter.

Response: This is a great point. We added the issue of family-wise error rate to the introduction. [Lines 16-21]

p.2: To introduce the idea of an unbalanced design, the authors consider the case of a linear regression with four equally spaced doses. The fact that the best design allocates half the observations to the smallest and largest dose and none to the intermediate ones does not mean the design is balanced. It's quite balanced between the highest and lowest dose. It is a well known fact that with linear regression, only two doses are needed to estimate the regression coefficient and that this is the optimal design. This design does rest on the strong assumption that there is no departure from linearity, and this is a very strong assumption that many practitioners may not be willing to make. This, I would hardly ever recommend such a design to a weed scientist in practice.

Response: This is also a great point. The example of the linear model was to help readers understand the concept of an experimental design (distribution of experimental units). Weed scientists may be (or should be) interested in an adequate strength of a treatment from a variety of perspectives such as the effectiveness of weed control, the impact on the environment, and the cost. In this regard, finding an adequate concentration (or any quantification of treatment strength) would be an important research objective. In a later section, we demonstrate a statistical model and an experimental design to find such a parameter in terms of delaying weed emergence. This point is added to the introduction of the revised manuscript. [Lines 67-76]

p.2, bottom: The authors use the term "two-phase design" for an approach that takes a first example to inform about the parameters of the model to be fitted and then in a second phase adapts the sample size to achieve a desire precision. I know at this point what the authors are intending but notice at the same time that the term "two-phase design" can have quite different meaning in other contexts. See, e.g.,

Brien, C. J. (2019). Multiphase experiments with at least one later laboratory phase. II. Nonorthogonal designs. Australian & New Zealand Journal of Statistics, 61:234-268.

I am more familiar with the term "adaptive design" for the approach the authors are pursing. These types of design are very common in clinical trials, and there is a very rich literature on this kind of design. The authors do not seem to be aware of this literature. Perhaps they can consider adopting this term to avoid confusion. Moreover, delving into the relevant literature may be useful. The authors do mention some work along these lines in a cursory manner at the very end of the paper, but it would be appropriate to state this from the very start, making it clear that the paper presents nothing that is in itself new and that much of the relevant theory was developed in the context of clinical trials, where one can find ample guidance for the types of design the authors are advocating.

Response: We appreciate your expertise on this. Throughout the paper, we replaced the “two-phase design” by the “adaptive design” and introduced the common use of adaptive designs in clinical trials early in the paper (in the introduction). [Lines 43-48] We made it clear that adaptive designs are not new in scientific communities, and it is discussed in the article in the context of agricultural studies. [Lines 97-100]

p.3: Having introduced linear regression as one example of an "unbalanced design" in the introduction, it is a bit surprising that Section 2 focuses on the comparison of two means. The connection with linear regression is not at all clear. Also, in the introduction the authors mention various examples of traits that involve counts and as such do not obey the normality or homogeneity assumption. Readers would expect that methods for count data are considered, e.g. generalized linear models. It's actually not clear after the introduction what will be the overall structure of the paper and how many different cases will be considered.

Response: You made a valid point. We did not intend to connect the comparison of two means with linear regression. To avoid such a confusion, we edited our language in the example of linear regression in the introduction (so it does not sound like a connected example). [Lines 57-62] In addition, we could not include all possible kinds of variable types in this article. We briefly mentioned the different nature of count data (non-normality and/or heterogeneity) and added some references related to experimental designs under GLMs. [Lines 148-151] Finally, we introduced the overall structure by the three parameters of interest discussed in the paper. [Lines 81-85]

p.3: Section 3 really lacks an agricultural example to motivate and illustrate the method. There is a large branch in agricultural sciences that relies primarily on field experiments with annual crops. In these settings there is rarely the luxury to use an adaptive design. The authors are still assuming normality here, whereas the introduction motivated the need to assume heterogeneity of variance using the example of count data. As pointed out before, there are tailored methods for count data, which would be prefered in this setting, and based on the introduction, this is what I would have expected in Section 2.

Response: We assume that this comment is on Section 2 (comparing two groups) not Section 3. Our intention was to start an example with the simplest case for diverse levels of readers. Some journals, reviewers, researchers, and practitioners believe that a conclusion is more convincing when results are reproduced (e.g., two seasons of data). As mentioned in the introduction, the impact of violating normality assumption mitigates as the sample size increases (e.g., a Poisson distribution gets close to a normal distribution), but a large sample size does not mitigate the violating homogeneity of variance. In large-sample studies, the homogeneity assumption is not required, and the two-sample t-test would work with unequal variance assumption. In this regard, researchers still can benefit from an adaptive design (if it is feasible) by estimating the unequal variances after the first phase of an experiment. We added a small section to address these points at the end of Section 2. [Section 2.3]

In Section 3, the authors consider a second example, i.e. a dose-response relationship where the interest is in finding the dose (denoted as Delta) that doubles the time to emergence of a weed. The authors suggest what they denote as a c-optimal design, but they give no reference whatsoever for this standard design problem. The assert that the information about Delta can be increased by minimizing the prior expectation of the variance of the parameter vector, but I am not seeing any proof of this claim. Moreover, what we really want is the optimal design for estimating Delta, not a design that "increases the information about Delta" (compared to what baseline design?). Finally, it is not even explained how the "optimal" number of replications are to be determined, let alone the number of x levels and their placement along the x-axis. All of this looks to be a standard design problem in nonlinear regression for which canned solutions are available but the authors make it seem like an entirely level problem. Showing in simulations that the equally replicated designs can be beaten is knocking a strawman. Incidentally, there is quite a lot of literature on time-to-event data as used in weed science, and there are several good R packages for this kind of data. The lognormal is but one distributional model for such data, and I am wondering why the authors picked this particular distribution. Perhaps because some of the algebra is straightforward? What if the times-to-event follow a Weibull or Gompertz distribution, for example?

Response: References for the c-optimal design have been added in the revised manuscript. [Lines 204-205] The c-optimal design is devised to minimize the expected asymptotic variance of the maximum likelihood estimator for the parameter of interest. It is what we meant by “increase the information about Delta,” but it was not an accurate language, so we corrected our language in the revised manuscript. In addition, we further explained (using calculus notation) how the optimal number of replications are to be determined (with an explanation using lay language). [Lines 205-211] There are many available experimental designs (not only c-optimal design) which can easily outperform the balanced (equally replicated) design, and we wanted to inform and demonstrate this fact to agricultural researchers who commonly choose the balanced design. [Lines 212-216] In addition, we added references for discussion of survival analysis in weed science. The choice of lognormal distribution is for mathematical convenience and practical interpretation [Lines 177-178].

Section 4.1: I have tried hard to understand the rationale of the computations leading to the prior for this example but failed. Section 4.2 only mentions right-censoring in passing, focussing on the derivation of a design under a right-censored model. Such a model was never mentioned in Section 3, so as a reader I am confused here and do not know what is going on. Why censoring all of a sudden? I think this kind of modelling needs a much more thorough introduction for the intended audience, which I think is agronomists and weed scientists. Censoring is a key property of such data, so just mentioning this in passing as if everybody knew this does not seem appropriate for the intended audience. Regarding the example, I note that the authors are assuming 10 seeds per pot. Each seed may have a different germination time, and it is most informative to analyse individual plant data. These data are clustered, however, and this would require fitting random pot effects.

Response: We appreciate your comment on the structure of paper. To explain the right-censored model and the prior specification more thoroughly, we restructured Section 4. In the revised manuscript, we first introduced the right-censored model under the log-normal assumption (Section 4.1) and prior specification (Section 4.2), then we introduced the applied example (Section 4.3) and retrospective simulations (Section 4.4). The experiment in Section 4.3 was a small-scale (pilot) study conducted in a balcony, and we tried hard to reduce random-effects due to soil characteristics, irrigation, and location. In these major changes, comments of the other reviewers are also addressed, and some typos are corrected. 

In a large-scale field experiment, as you pointed out, we believe that substantial random-effects would exist due to the clustering, and it should be considered via a more complex model such as a mixed-effect model. It is beyond the scope of this article, and it is discussed at the end of Section 4 with references related to optimal designs under mixed-effects models (Section 4.5). [Section 4 is substantially revised and restructured.]

Section 5 considers a logistic regression model to assess synergistic effects between two quantitative inputs. The response is binary, and germination of individual seeds is considered as a case in kind. This is just a hypothetical example, no real experiment is considered. The main challenge with this kind of data, again, is clustering. Several seeds are usually placed on the same experimental units (pot, petri dish, filter paper), and the number of germinated counts is assessed per experimental unit. These binomial counts are very typically overdispersed, which is the main challenge in the analysis. The whole derivation in this section ignores overdispersion, so I am afraid it is not relevant for most practical purposes.

Response: Your point is well taken, and it is beyond the scope of this article. We added sentences to alert the importance of over-dispersion and relevant references at the end of Section 5. Thank you for your expertise and valuable feedback. [Lines 397-404]

In summary, my impression is that the authors do not have a very intimate connection with experimenters and real experiments in the weed sciences and that the methods they propose are not very practical. What I find mainly lacking are convincing real examples that fully motivate the proposed methods. Most of the derivations are based on rather hypothetical and artificial settings.

I do applaud the general intention of the paper, which I would say is focused on the optimal design for regression models in weed science. I believe, however, that this work would benefit greatly from a closer collaboration with weed scientists. If examples can be presented where the proposed designs have really been found to work for weed scientists, this could make a much more convincing case. This may also mean rather different designs presented as desirable in the end.

Response: We understand that there is a gap between the real experiments and simulation/example provided in this article. At the same time, we see a gap between current practice (e.g., balanced designs) and general understanding of experimental designs from statistical perspective (even under the simplified hypothetical situations). The primary point of this article is to demonstrate the importance of experimental designs (breaking the misconception of the superiority of the balanced design) and adaptive designs (i.e., the entire experimental units do not have to be spent at once, and adaptive decisions can improve the precision of parameter estimation). We greatly appreciate your expertized perspective, and most of your feedback is addressed by noting readers with references. Thank you.

Reviewer #3: The study is focused on one of the most important issues in weed science, experimental designs. The concept structured was firstly described and given its theoretical background. Although the data presented by the authors to clarify their ideas was enough, it would be obtained more realistic and powerful from herbicides instead of EtOH. Under different scenarios, the two-phase design may create lower MSE’s than single stage design.

Response: Thank you for your valuable perspective. We agree that there are more realistic and powerful herbicides in weed science. The experiment of EtOH was conducted due to our restricted budget and logistics (during the pandemic), and it was used for the purpose of demonstrating the experimental design and parameter estimation. In the revised manuscript, this point was made clear in Section 4.3. [Lines 280-282]

Please check the sentence in line 80-81 “The null hypothesis is H0: µ1- µ 2 = 0, and the alternative hypothesis is H1: µ1- µ 2 = 0”.

Response: Thank you for catching the typo. We meant H1: H1: µ1- µ2 ≠ 0. It is corrected. [Line 107]

Please check the value in line 216 “x3 = 0.5”.

Response: Thank you for catching the typo. We meant x4 = 0.5. It is corrected, and associated typos are also corrected. [Lines 296, 298, and 306]

Please add separate captions for each figure.

Response: Thank you for your suggestion which is helpful for readers. We added separate captions (explanations) for each figure. [See captions of Fig 1-8.]

---

## [Decision Letter · Decision Letter 1]

16 Aug 2021

PONE-D-21-17452R1

Applications of statistical experimental designs to improve statistical inference in weed management

PLOS ONE

Dear Dr. Kim,

Thank you for submitting your manuscript to PLOS ONE. After careful consideration, we feel that it has merit but does not fully meet PLOS ONE’s publication criteria as it currently stands. Therefore, we invite you to submit a revised version of the manuscript that addresses the points raised during the review process.

Your last version is pretty good. I am looking for some additional changes which will improve the quality of paper. I guess it will be improved to be published quality.

We look forward to receiving your revised manuscript.

Kind regards,

Ahmet Uludag, Ph.D.

Academic Editor

PLOS ONE

Journal Requirements:

Additional Editor Comments (if provided):

I would like to accelerate your paper publishing. If you make relevant changes that are from the third referee and prepare and add a detailed rebuttal, I will check it myself after corrections. I think suggestions are very clear.

Reviewers' comments:

Reviewer's Responses to Questions

**Comments to the Author**

1. If the authors have adequately addressed your comments raised in a previous round of review and you feel that this manuscript is now acceptable for publication, you may indicate that here to bypass the “Comments to the Author” section, enter your conflict of interest statement in the “Confidential to Editor” section, and submit your "Accept" recommendation.

Reviewer #1: All comments have been addressed

Reviewer #3: All comments have been addressed

Reviewer #4: All comments have been addressed

2. Is the manuscript technically sound, and do the data support the conclusions?

Reviewer #1: Yes

Reviewer #3: Yes

Reviewer #4: Partly

3. Has the statistical analysis been performed appropriately and rigorously? 

Reviewer #1: Yes

Reviewer #3: Yes

Reviewer #4: No

4. Have the authors made all data underlying the findings in their manuscript fully available?

Reviewer #1: Yes

Reviewer #3: Yes

Reviewer #4: Yes

5. Is the manuscript presented in an intelligible fashion and written in standard English?

Reviewer #1: Yes

Reviewer #3: Yes

Reviewer #4: Yes

6. Review Comments to the Author

Reviewer #1: (No Response)

Reviewer #3: (No Response)

Reviewer #4: Article: Applications of statistical experimental designs to improve statistical inference in weed management

General Comment

In addition to the weed branch, a useful study has been written that will easily adapt to the agricultural field. The article is written from an innovative perspective. Although the study contains intensive information theoretically, there are doubts about its transfer to practice. The number of iterations (1000) of the simulation study was kept at an insufficient level. It is known in the literature that the results change with the increase in the number of simulations. Therefore, the number of iterations should be increased in the simulation. The simulation steps can be made more understandable by showing them with a diagram. The distributions used in the simulation study do not include all the conditions in terms of the shape of the distribution. Therefore, especially t(10), t(5), β(5,10), β(10,5) and χ2(3) distributions should be included in the study. In addition, while generating random numbers, they should be produced and reported by taking into account different effect sizes. As a result of the study, could different criteria be used besides Bias, Variance, and MSE arguments? (Table 1 and Table 2). So, are these criteria a sufficient argument for comparing the results?. Comparing the power (β) and type I error rates (α) of each of the tests in the scenarios created will strengthen the study. Which packages of the R program were used for the simulations?. If the package is not used and a new function is written, it should be stated in the manuscript.

Specific comments

Abstract

This part is written very generally and the material of the study, the methods applied and the results should be mentioned. The author needs to re-write this and improve novelty.

Introduction

Line 16 – 17, The criticism made for Duncan multiple comparison does not reflect the truth. In recent simulation studies, Duncan's superiority over other methods is understood. These statements should be supported by the literature.

Line 22, the sentences are not flowing well in paragraph entry. Please rephrase

Line 22 and 25, These statements should be supported by the literature.

Line 30 and 33, These statements should be supported by the literature.

Line 43 and 44, First, why wasn't the sample size determined using power analysis instead? Second, instead of dividing the dataset and estimating group variances, why not crossvalidation by dividing the dataset into testing and training?

The differences between the two-stage design and the bayesian approach should be clearly stated.

Materials and Methods and Results

This section is difficult to follow especially the results, are not well structured and clearly reported.

Only the normal distribution was used in the simulations for two-group comparisons. It is known that the shape of the distribution affects the power of the test. Therefore, the distributions N(0,1), t(10), t(5), β(5,10), β(10,5) and χ2(3) should be used.

The simulation steps can be made more understandable by showing them with a schematic diagram.

It is useful to give the power of the test in tables as well as graphs.

Discussion

Discussion of the results of the study in this section is rather poorly. It's more like the conclusion part. This section needs a review as there are only 8 literatures in this section.

Conclusion

This part does not exist. What is the take home message from the current study? This should be addressing the main aim (also the topic).

Reference

T ASLAN, E., KOŞKAN, Ö., & ALTAY, Y. (2021). Determination of the Sample Size on Different Independent K Group Comparisons by Power Analysis. Türkiye Tarımsal Araştırmalar Dergisi, 8(1), 34-41.

7. PLOS authors have the option to publish the peer review history of their article (what does this mean?). If published, this will include your full peer review and any attached files.

Reviewer #1: No

Reviewer #3: **Yes: **Ahmet Tansel Serim

Reviewer #4: No

---

## [Author Response · Author response to Decision Letter 1]

21 Aug 2021

We greatly appreciate the opportunity to improve our manuscript. We responded to all comments made by the reviewer, and it is attached as a separate PDF file in our submission. Thank you.

---

## [Decision Letter · Decision Letter 2]

2 Sep 2021

Applications of statistical experimental designs to improve statistical inference in weed management

PONE-D-21-17452R2

Dear Dr. Kim,

We’re pleased to inform you that your manuscript has been judged scientifically suitable for publication and will be formally accepted for publication once it meets all outstanding technical requirements.

Kind regards,

Ahmet Uludag, Ph.D.

Academic Editor

PLOS ONE

Additional Editor Comments (optional):

Congratulations. But, check some minor problems as reviewer pointed.

Reviewers' comments:

Reviewer's Responses to Questions

**Comments to the Author**

1. If the authors have adequately addressed your comments raised in a previous round of review and you feel that this manuscript is now acceptable for publication, you may indicate that here to bypass the “Comments to the Author” section, enter your conflict of interest statement in the “Confidential to Editor” section, and submit your "Accept" recommendation.

Reviewer #4: All comments have been addressed

2. Is the manuscript technically sound, and do the data support the conclusions?

Reviewer #4: Yes

3. Has the statistical analysis been performed appropriately and rigorously? 

Reviewer #4: Yes

4. Have the authors made all data underlying the findings in their manuscript fully available?

Reviewer #4: Yes

5. Is the manuscript presented in an intelligible fashion and written in standard English?

Reviewer #4: Yes

6. Review Comments to the Author

Reviewer #4: Please consider the following minor corrections.

In Line 161) " t(10), t(5), β(5, 10), β(10, 5) " should be used instead of " T(10), T(5), Beta(5, 10), Beta(10, 5".

In Line 361 Is the number of iterations 1000 or 10000? Please check?

7. PLOS authors have the option to publish the peer review history of their article (what does this mean?). If published, this will include your full peer review and any attached files.

Reviewer #4: No

---

## [Editor Report · Acceptance letter]

6 Sep 2021

PONE-D-21-17452R2 

Applications of statistical experimental designs to improve statistical inference in weed management  

Dear Dr. Kim:

I'm pleased to inform you that your manuscript has been deemed suitable for publication in PLOS ONE. Congratulations! Your manuscript is now with our production department. 

Kind regards, 

on behalf of

Dr. Ahmet Uludag 

Academic Editor

PLOS ONE